# A Cross Stage Partial Network with Strengthen Matching Detector for Remote Sensing Object Detection

**Shougang Ren, Zhiruo Fang and Xingjian Gu ***

College of Artificial Intelligence, Nanjing Agricultural University, Nanjing 210095, China
* Correspondence: guxingjian@njau.edu.cn

**Abstract:** Remote sensing object detection is a difficult task because it often requires real-time feedback through numerous objects in complex environments. In object detection, Feature Pyramids Networks (FPN) have been widely used for better representations based on a multi-scale problem. However, the multiple level features cause detectors' structures to be complex and makes redundant calculations that slow down the detector. This paper uses a single-layer feature to make the detection lightweight and accurate without relying on Feature Pyramid Structures. We proposed a method called the Cross Stage Partial Strengthen Matching Detector (StrMCsDet). The StrMCsDet generates a single-level feature map architecture in the backbone with a cross stage partial network. To provide an alternative way of replacing the traditional feature pyramid, a multi-scale encoder was designed to compensate the receptive field limitation. Additionally, a stronger matching strategy was proposed to make sure that various scale anchors may be equally matched. The StrMCsDet is different from the conventional full pyramid structure and fully exploits the feature map which deals with a multi-scale encoder. Methods achieved both comparable precision and speed for practical applications. Experiments conducted on the DIOR dataset and the NWPU-VHR-10 dataset achieved 65.6 and 73.5 mAP on 1080 Ti, respectively, which can match the performance of state-of-the-art works. Moreover, StrMCsDet requires less computation and achieved 38.5 FPS on the DIOR dataset.

**Keywords:** object detection; one-stage detector; multi-scale; StrMCsDet

## 1. Introduction

Object detection technology has rapidly developed due to the development of technology. Traditional object detection methods extract features from candidate regions in images and then classify them using Support Vector Machine (SVM) [1] models. Nowadays, instead of manual feature extraction, deep learning-based methods have automatically been used as learning image features. Therefore, most object detection methods for the remote sensing field are based on deep learning methods in recent years. However, in real world tasks, mainstream detection methods have strict constraints under real conditions, including hardware performance, detection efficiency, and accuracy. Hence, effective implementation in real world detection is needed and is necessary to propose an effective detection model for remote sensing images.

Object detection methods can be divided into two categories, including two-stage and one-stage object detection. Two-stage methods first extract the candidate region of the target and then classify the objects in the region. R-CNN series methods including R-CNN [2], Fast-RCNN [3], Faster-RCNN [4], and Mask-RCNN [5] use the selective search to extract all candidate regions in advance, improving the Convolutional Neural Network (CNN) practical. One of the advantages of these methods is the high precision of detection. In two-stage methods, the extract RoI feature falls behind with the detector extract multiple features. R-CNN [2] and Fast-RCNN [3] are limited by selective search algorithm. By using two-stage decoders, the accuracy of detection has been greatly improved and many improved models have been proposed in recent years. Most successful detectors are based

on the R-CNN framework. However, detection speed is also important in practical tasks. A two-stage detector cannot satisfy real-time performance, so it is a trend to design a faster detector.

One-stage methods are proposed to speed up object detection. Seeing detection as a regression problem, a one-stage detector is faster than two-stage methods. For the one-stage method, the network performs localization and classification without a proposal box, which speeds up the detector. The Single Shot Detection [6] (SSD) network, as a one-stage detector, using VGG-16 as a backbone to extract features, predicts each feature layer of different scales. In the same year, You Only Look Once [7] (YOLO) proposed seem object detection as a regression problem and greatly enhanced the speed. YOLO series including YOLOv1 [7], YOLOv2 (YOLO9000) [8], YOLOv3 [9], YOLOv4 [10], YOLOv5 and many other related works [11–13] have made great contributions in the field of real-time object detection. However, imbalance between the positive and negative object problem greatly affects the performance of one-stage detectors. To solve the problem, a work that combines the ResNet-101 [14] classification network and Feature Pyramid Network to achieve multi-scale detection, RetinaNet [15], proposes focal loss to classify imbalance. The speed of the one-stage method is generally faster than a two-stage detector, but accuracy is lost.

With the improvement of accuracy in object detection frameworks in recent years, train and inference speed are simultaneously important. Feature pyramids [16] become an essential component in most methods. In YOLOF [17], the author studies the essential factor of the success of the Feature Pyramid Network. The Feature Pyramid Network provides a solution of conquer and divide. However, the pyramid structure slows down the detector and brings forward memory burden because of the multi-level structure. Feature pyramids are used to fuse multiple feature inputs, including low-resolution and high-resolution features. Multi-scale problems involve locating and classifying targets at different scales. To fuse, multiple level feature is one of the solutions; in detectors with feature pyramid, they construct multiple-level features with receptive fields which match with different scales. Many works optimize feature fusion strategy to speed up the detector. Anchor-free methods, such as CornerNet [18] and CenterNet [19], that only use the last level feature can be fast and detect all the objects on an one-level feature. DETR [20] adopts a transformer [21]. Vision transformers [22] also use transformers to detect tasks and only use a single C5 feature but they need a long training schedule for convergence. Unlike these works, StrMcDet in this paper provides an alternative solution to replace feature pyramids.

Additionally, object detection of the remote sensing image is different from that of the natural image, including object class imbalances, inter-class diversity and inter-class similarities, and a larger field of view. In size, the remote sensing images are much larger than natural images. Meanwhile, the complex scene contains more messages that are similar to each other, such as 'bridge' and 'dam', 'overpass' and 'bridge', etc. (Figure 1). In real scenarios, not only is accuracy of importance, but also efficiency, according to the real-time tasks that many researchers [23–26] turned to.

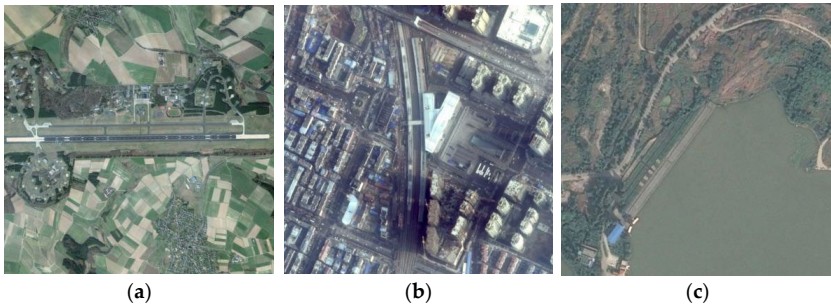

|  (a)  |  (b)  |  (c)  |

**Figure 1.** Remote sensing images in DIOR dataset, (**a**) contains 'Bridge', (**b**) contains 'Overpass', (**c**) contains 'Dam', which have similarities with other classes.

Considering the above problems, this paper proposes an efficient one-stage detector for remote sensing images. Contributions of this paper are:

1.　This paper proposes a one-stage detector which uses single-feature layers instead of a traditional feature pyramid. The output of the backbone is the only C5 feature map, which introduces an alternative way to the traditional feature pyramid;
2.　This paper proposes an encoder with residual blocks containing a dilated attention module. To compensate, the receptive filed is absent due to the single-level feature output in backbone. The response of the detected object is enhanced with the decoder and the high resolution of feature maps are maintained;
3.　To balance the positive and negative objects of different scales, this paper proposes a strategy called strengthen matching method. Positive anchors append to the nearest target for a better match to the original ground truth. Objects of different scales can be equally matched due to this strategy.

The method replaces the complex pyramid by single-level feature output. However, the replacement causes a receptive field limitation. To solve the problem, the dilated encoder is one of the key components of StrMCsDet. Another key component is the strengthen matching strategy which ensures accurate results.

Experiments on DIOR [27] and NWPU-VHR-10 [28] both achieve a comparable result in both accuracy and velocity. The baseline is an effective method under real scenarios for remote sensing image detection tasks.

This paper is organized as follows. Related works about this method are mentioned in part 2. The proposed StrMCsDet is introduced in detail in part 3. Part 4 introduces datasets of the experiments. Part 5 provides experiments and discussions. We give an analysis in part 6. Finally, we conclude this article and plan for future work in part 7.

## 2. Related Work

Convolutional neural networks (CNN) have attracted a lot of attention in the field of computer vision. To speed up detection, a one-stage detector was proposed. You Look Only Once (YOLO) series including YOLOv1 [7], YOLOv2(9000) [8], YOLOv3 [9], and subsequent modified YOLO methods convert the object detection work into a regression problem, the method utilizes the bounding box locations, and their classes are obtained by using the whole image as the input of the bounding box. Subsequent works adopt the backbone of the CSPNet [29] to enhance the learning ability of CNNs. YOLO series have a fast detection speed and can process 45 frames per second, making it easy to deal with real-time tasks. In this paper, we adopt a one-stage baseline to deal with real world tasks. The MobileNet [30] aims to construct a lightweight object detection algorithm. The MobileNet [30] was designed for mobile and embedded visual applications. Detection work based on MobileNet [31] focuses more on the lightweight of the overall model. For segmentation, Ref. [32] we adopted the MobileNet as the backbone which achieved high performance in mineral image tasks. In the real task, the natural scene contains a large number of pictures [33] and considers using the generation of an adversarial network to reduce the cost consumption of detection. Nowadays, in image tasks, efficiency and model lightweight are problems worth exploring.

In recent years, Feature Pyramid Network (FPN) has been widely used for feature fusion. For remote sensing detection, Ref. [34] we deal with the ship using improved YOLOV5 with FPN. In IRFR-Net [35], the weighted heterogeneous space is introduced to obtain a multi-scale feature. In RMCHN [36], convolution layers combined with long and short path feature learning strategy were used to fuse captured features and improve feature representation ability. The Feature Pyramid Structure was also used in 3D works, as in 3D cloud object segmentation [37,38] for overhead catenary height detection. The backbone of RetinaNet [15] is similar to FPN [16]. In popular detection work, FCOS [39] also applied this structure. However, unlike FPN, which uses C2, RetinaNet [15] does not, because P2 generated by C2 takes up more computational resources, so the authors produce P3 directly from C3; as a result, each feature map is represented by a single layer

of the backbone. In YOLOF [17], the author also analyzed the success of feature pyramid and proposed a simple baseline to avoid a complex structure. In that, to create a network that combines accuracy and efficiency, there does not necessarily need to be a complete pyramid structure used for detection. The M2Det [40] and the CF2PN [41], the improved pyramid model, is used to improve detection accuracy for multi-category remote sensing data.

In the one-stage detector, classification and regression in one branch takes the imbalance problem. The Focal Loss [15] function aims to solve the problem of positive and negative sample imbalance. It focuses on adding weight to the loss corresponding to the sample according to the difficulty of sample discrimination, which effectively alleviates the problem of sample imbalance. RetinaNet, which uses focal loss as the one-stage network, surpasses the two-stage network for the first time in 2017.

In addition, attention mechanisms are widely used in recent visual works. The attention mechanism was first used in natural language processing (NLP), which is a kind of deep learning optimization strategy from human attention. In [42], a novel and independent module based on the attention mechanism was proposed, which can be embedded into classification and detection work for detection tasks. The convolutional block attention module [43] (CBAM) considers the imbalance problem in both spatial dimension and channel dimension. In the CBAM, the channel attention map and spatial attention map area adopted to utilize the inter-channel and inter-spatial relationship, respectively. With both max-pooling and avg-pooling in two dimensions, it can extract abundant high-level features. In SA-FPN [44], the FPN is equipped with multi-scale feature fusion and an attention mechanism to improve human detection performance in real world tasks for a crowd scenario.

In this paper, we propose a one-stage detector for remote sensing image object detection. The method adopts the backbone of a cross stage partial network to reduce the computation burden and build a lightweight model. The output is a single-level feature which contains enough context for detection but is limited in scale. Thus, an improved residual block with attention module deals with the multi-scale problem on a single-level feature map. Then, a strengthen matching method based on enhance anchor matching is used to prevent the influence of extreme size objects. The supplement generates positive samples to dynamically match the ground truth box under a certain offset, which is better only for using anchor and ground-truth matching.

## 3. Proposed Method

### 3.1. Framework Overview

The major goal of StrMCsDet is to achieve efficiency alongside accuracy in remote sensing object detection tasks. The StrMCsDet consists of the backbone, the encoder, and the decoder. The overview of the architecture is shown in Figure 2. The backbone consists of cross stage partial layers to achieve richer gradient combinations and reduce the computation burden. For detection of numerous targets, the single C5 feature contains enough context. To compensate for the lack of receptive field of the single-level feature, we lined up eight residual blocks in the encoder. Residual blocks compensate for the receptive field limitation with a dilatation module of eight different scales. The decoder asymmetrically consists of the classification head and regression head. The classification head contains two convolutions followed by batch normalization layers and ReLU layers. Four convolutions make up the regression head. For the regression head, an implicit objectness prediction without direct supervision adds to generation of the final score of prediction. The objectness branch is parallel with the regression head and is multiplied with the classification head, which is used to suppress a high response in the background region.

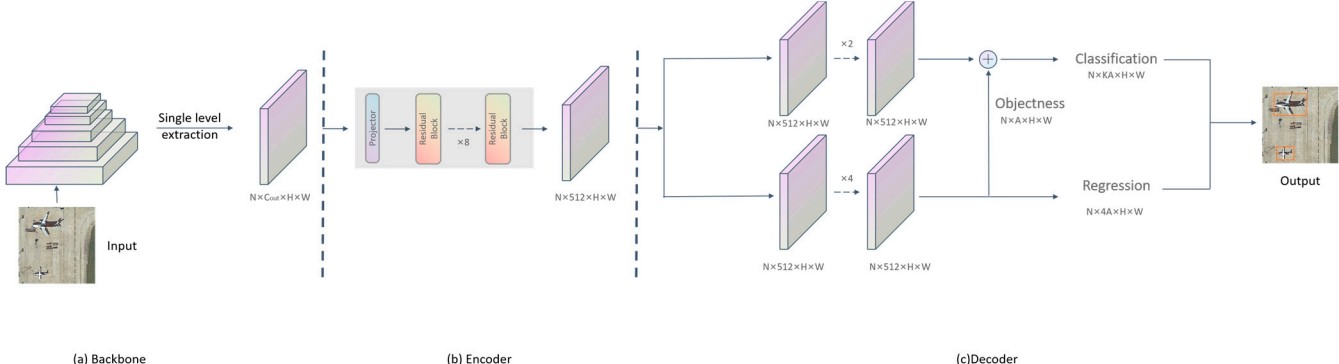

**Figure 2.** Overall framework of StrMCsDet.

### 3.2. Single-Level Feature Extraction with Cross Stage Partial Backbone

The basic backbone for feature extraction of StrMCsDet is shown in Figure 3. At first, for the input after completion of the convolution, batch normalization is normalized with the activation function Mish. Mish has a low cost and smooth, non-monotonic, upper-unbounded, lower-bounded features that improve its performance compared with other commonly used functions, such as ReLU. Then, the resblock body module is stacked, which consists of one downsampling and multiple residuals stacked. In each resblock body, we adopted cross stage partial architecture. The cross stage partial block was used instead of the residual block in the network, as shown in Figure 3b. The CBM block contains a convolution layer, a batchnorm layer, and a Mish layer. There are total n cross stage partial layers (CSP) in each resblock body. Each block consists of a downsampling layer and several stacked residual blocks with residual edges. The basic input size is w × h × c. In StrMCsDet, the image size is 608 × 608 × 3. The output single-level feature map is sized as 19 × 19 × 1024.

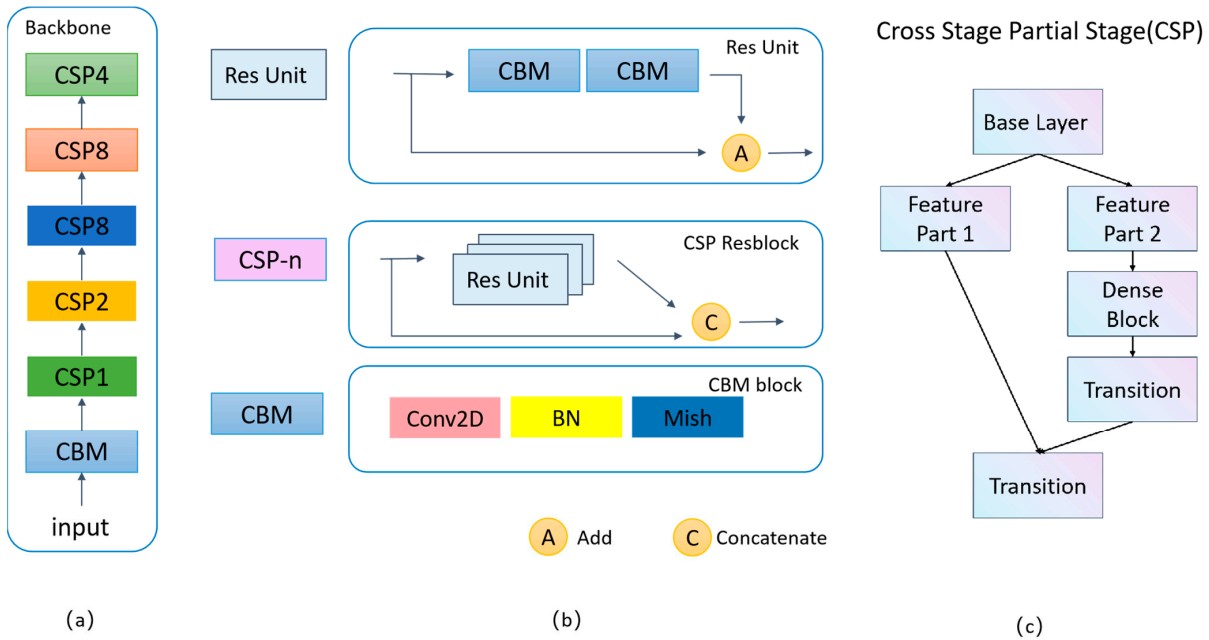

**Figure 3.** (**a**) Backbone of the StrMCsDet. (**b**) Details of each block. (**c**) The feature fusion strategy in the cross stage partial layer is partial dense block and a transition block is contained in the block, transition at the first and then concatenation and transition at the last.

Neural networks are powerful when they become wider and deeper. Such as in object detection, deeper networks take more computation which makes for heavy tasks. In

CSPNet [29], the author found that one of the reasons for the large computation was that the gradient information was repeated in the process of network optimization. If the repetition of gradient information can be effectively reduced, the learning ability of the network will be greatly improved. Redundant gradient information problems in the backbone can result in costly inference computations and inefficient optimization. A large amount of gradient information is repeatedly used to update the weights of different dense blocks. This will cause different dense blocks to repeatedly learn the same gradient information. As shown in Figure 3c, in the backbone, we adopted the cross stage partial module in the resblock body. The main part continued to stack the original residual blocks. The other part is like a residual edge, with a small amount of processing directly connected to the end. The cross stage partial block module makes the gradient flow into two different paths to increase the correlation difference of gradient information. Compared with the residual block module, the cross stage partial block module adopted in the backbone can enhance the learning ability of the convolution network and improve its accuracy. The backbone utilizes fewer layers of output to reduce computation burden. The effectiveness of this design is demonstrated by experiments in part 5. The backbone effectively avoids the model turning into a minimum value problem in the iterative process to accelerate the convergence of training. Transition is an idea in which the layers separate features into two parts; the method can be combined with multiple networks.

The StrMCsDet utilizes single-level feature output. Output of the backbone is only in the C5 feature map. In traditional Feature Pyramid Networks (FPN), as shown in Figure 4a, the pyramid improves the detector's accuracy by fusing features on different scales. The feature pyramid's feature map's concatenation, however, is absolutely enormous. In YOLOF [17], the author proves that the single-in-single-out method can achieve a comparable result with that of the multiple-in-multiple-out method. The performance gap between multiple-in-, multiple -out, and multiple -in-single-out is less than 1 mAP, which proves that C5 carries enough context for detection.

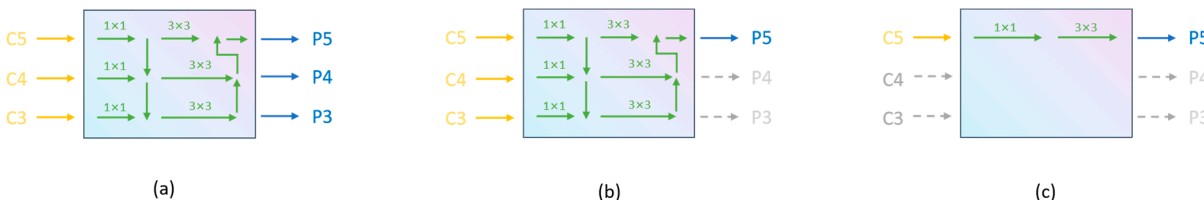

**Figure 4.** (**a**) Traditional feature pyramid multiple-in-multiple-out structure, (**b**) multiple-in-single-out structure, (**c**) single-in-single-out structure in this method.

FPN takes calculation burdens and make detectors slow. In a one-stage detector like RetinaNet, multiple level feature makes the structure complex. As the C5 feature for detection contains enough context, in our method, we try a single-level feature output with the cross stage partial backbone. It shows a simple way to reduce memory burden.

In StrMCsDet, we adopted a single-level feature output. However, fewer layers of features can cause a problem with the limitation of receptive fields. To solve this problem, we designed an encoder to subsequently deal with and improve the performance of the detector.

### 3.3. Encoder with Stacked Multi-Scale Residual Blocks

The StrMCsDet is a single-in-single-out detector. Single layers influence reference speed but they also address the issue of scale limitations. The receptive field covered in C5 feature is constant in a limited range, which results in scale limits with part of the object. To detect all scale objects, we proposed an encoder with dilation blocks to generate an output feature with different receptive fields and make up for the lack of limited features to detect objects of all scales. In StrMCsDet, we designed a component as an encoder to replace the pyramid structure.

Limited receptive fields leads to poor performance of detection. As shown in Figure 5a, the first example indicates that the original receptive field covers a limited scale range. Figure 5b is an enlarged receptive field which can cover a larger range but can miss some small objects. In the residual block, standard and dilated convolutions enlarge the receptive field of the C5 feature. However, the whole scale range shifts to larger scales as Figure 5b has shown. Then, we combine the range of multiple receptive fields by adding the corresponding features. With multiple receptive field scales, the whole range can be covered by the feature. As a result, the dilated encoder covered all object scales through stacked, successive, residual blocks. In the ablation experiment, we adapted different enlarged scales and adapted the hyper-parameter of 8 in remote sensing object detection. The dilated encoder can replace the traditional feature pyramid to generate a multi-scale feature. Additionally, it may keep more details in the output of the backbone and maintain the spatial resolution of the feature.

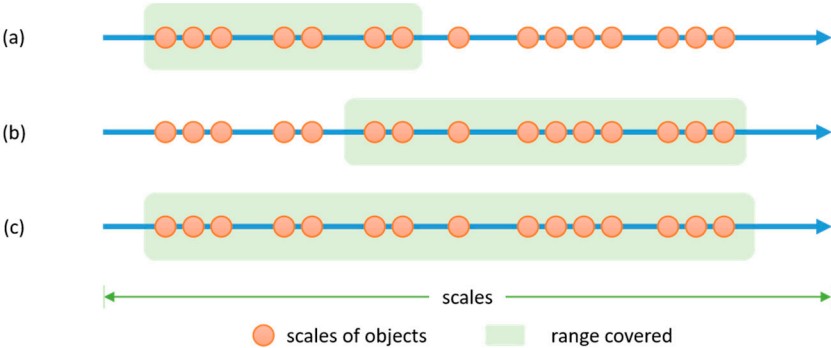

**Figure 5.** An example to illustrate the relation of the single-feature between the object scales and the scale range covered. Circle number represents the scale of object across the scale, the more the circle, the larger the scale. (**a**) the original receptive field, (**b**) dilated receptive field shifts to larger scale, (**c**) all the scales covered by stacked residual blocks.

Based on the above, the encoder contains four residual blocks with dilation. As shown in Figure 6, Figure 6a shows the structure of a projector and single residual block. The first layer is a projector which uses a $1 \times 1$ convolution layer to change the channel dimension. As the function of feature pyramids, a $3 \times 3$ convolution layer to generate output has its setting dilated to cover all scales. The residual blocks generate multiple receptive fields of feature from the backbone. The encoder with residual blocks one-by-one, instead of the feature pyramid, enables the detector to detect on different scales. As Figure 6b shows, the output maintains the resolution of the feature and keeps more details with the Convolutional Block Attention Module (CBAM) after the feature map has been dilated.

In residual blocks, we adopted a light attention module to enhance the encoder. The module was added into a $3 \times 3$ convolution layer with dilated block. Attention is an optimization strategy which imitates human attention when looking at an image in the visual field. For remote sensing images, using an attention module can effectively reduce the interference of external environments. To control the information transmitted into the deep feature extraction modules, we added a simple attention module for feed-forward convolutional neural networks. The Convolutional Block Attention Module (CBAM) includes two sequential sub-modules, channel and spatial. In this paper, we added the module in each residual block together with dilated blocks. For the channel attention module, the channel dimension remains unchanged and focuses on object classification. For the spatial attention module, the spatial dimension remains unchanged and focuses on the target location. In remote sensing image vision tasks, this added module can be used to focus more on the object itself, improve the precision, and remove redundant information.

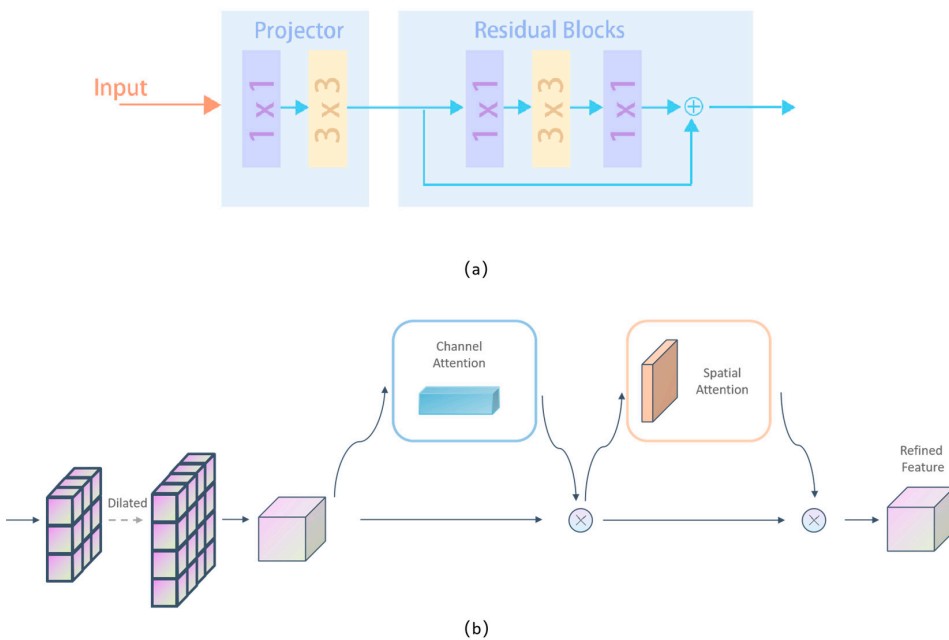

**Figure 6.** The structure of encoder. (**a**) $1 \times 1$ and $3 \times 3$ means the convolution layers, in StrMCsDet, with eight successive residual blocks with dilated and attention module. (**b**) The $3 \times 3$ convolution layer includes dilated block, then the channel and spatial attention module to enhance the feature map.

The encoder deals with the scale problem and receptive field problem, and the residual one-by-one blocks share the same weight. This keeps more of the original feature's details while maintaining the resolution of the feature. The spatial and channel attention mechanism puts more focus on the object itself. We generated a feature map with multiple receptive fields to enable object detection in a single-level feature map. In the encoder part, dilated residual blocks replace the traditional feature pyramid structure. Although the encoder is simple, it reduces computation and model size; detailed data are in part 5.3. Additionally, the dilated encoder covers different sizes of object from small to large because of the dilated module. The attention module enhances the detection effect from channel and spatial. The lightweight attention block focuses together on classification and location to improve precision. The dilated encoder is an essential part of single-level output in order to utilize the feature map and enhance the effect of the detector without redundant computation. The dilated encoder with the attention module deals with multi-scale instead of complex pyramids.

In the encoder part, to compensate for the lack of multiple-level features, we designed a stacked residual block to generate an output feature with various receptive fields on the single-level feature map. The combined original scale range and the enlarged scale range resulted in a feature with multiple receptive fields. The encoder works with our single-layer feature output and replaces conventional FPN. Instead of using multiple-level features, the suggested encoder enables us to recognize objects at all scales on a single-level feature. In Part 5, experiments were conducted to demonstrate the stacked dilated encoder's ability to deal with multi-scale problems and improve precision.

### 3.4. Strengthen Matching Method and Decoder

In this paper, we adopted fewer feature layers before, so that the number of the anchors would be less than the rounded pyramid structure. From 100 k to 5 k anchors, the anchor amount will have a large decrease in the fewer feature layer method. However, in real state tasks, the ground truth of large-scale boxes induces more positive anchors than small ones, which can lead to an imbalance problem between positive and negative anchors. During training, this problem will lead to the detector ignoring small ground truth boxes

while paying more attention to large ground truth boxes. In order to address the imbalance between large and small ground truth boxes matching for the decoder and for the better match of objects of different scales, we propose a strengthen matching strategy.

Figure 7 shows the distribution of the generated positive anchors in Max-IoU and our method. The histogram aims to show the balance of anchor generation. Imbalance problems in positive anchors are a challenge for detection tasks. In object detection, to correctly define a positive anchor is of significance. In anchor-based methods, the IoUs between ground truth boxes and anchors determine the positive. In RetinaNet, a threshold was set as 0.5, which means that when the max IoU of the ground truth and the anchor is more than the threshold, the anchor will be positive. For multiple levels encoders, the anchors are previously defined while ground truth boxes generate the positive anchor for different scales in the feature level. When there is a category imbalance in the data, the scale of the anchor box generated by the clustering algorithm will be biased towards the majority of categories, which makes detection performance of the minority categories poor. For RetinaNet, anchors are generated with areas such as $\{32^2, 64^2, 128^2, 256^2, 512^2\}$ in different layers (P3–P7). The anchor size is $\{2^0, 2^{1/3}, 2^{1/2}\}$ in each layer and the aspect ratio is $\{0.5, 1, 2\}$. In StrMCsDet, as a single-level feature output, with a one level feature to generate anchors, we placed 5 anchors in each position, anchor size is 1 and aspect ratio is 1. As an experimental result, larger anchor size and more aspect ratio does not affect the performance of StrMCsDet. Table 1 shows details of RetinaNet and StrMCsDet.

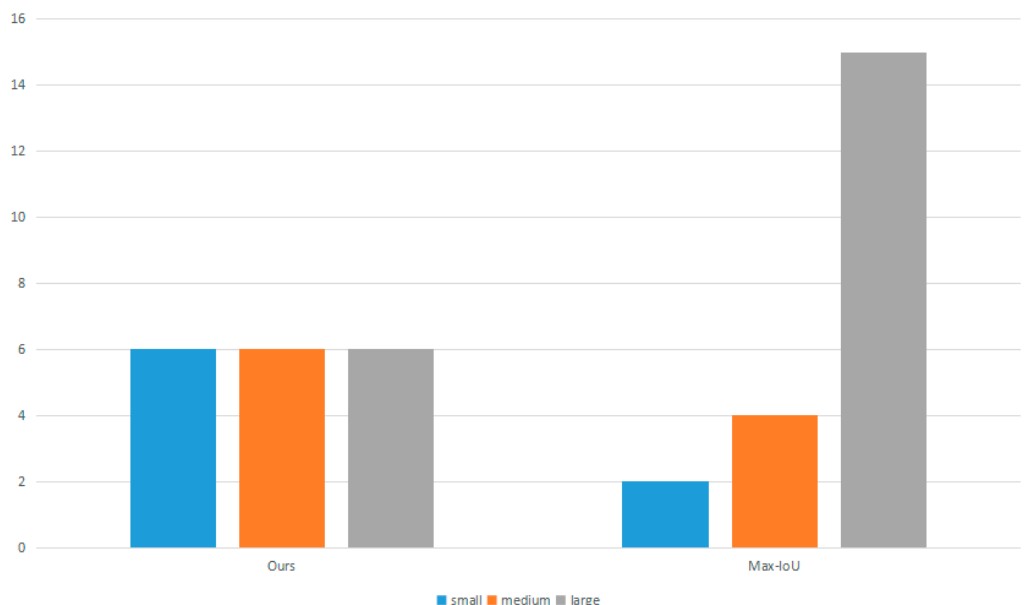

**Figure 7.** Positive anchors in Strengthen Matching and Max-IoU method. Large ground truth boxes affect the Max-IoU cause imbalance. Ours tend to match different scales equally with the box matching method.

**Table 1.** Anchor size and aspect ratio of RetinaNet and StrMCsDet.

| Model | Anchor Size | Aspect Ratio |
|---|---|---|
| RetinaNet | $\{2^0, 2^{1/3}, 2^{1/2}\}$ | $\{0.5, 1, 2\}$ |
| StrMCsDet | $\{1\}$ | $\{1\}$ |

The StrMCsDet collapses multiple anchors to fewer levels with different sizes of 32, 64, 128, 256, and 512 on each position of the feature layer. To relieve the imbalance problem, we enhanced the matching for ground truth boxes. For each ground truth box, we adopted the *k* amount nearest to the anchor for it to be positive, which causes each ground truth

box to be fairly matched with the same number of anchors, which has no connection with large or small scales, so that it is ensured that every sample contributes equally in the training process. The Max-IoU that we set in the strengthen matching ignores large IoU > 0.7 and small IoU < 0.15 positive anchors. A certain number of positive samples were supplemented and some ignored samples were also considered.

The decoder consists of the regression head and classification head as shown in Figure 8. The convolution layers' numbers of the two heads are different. The classification head consists of four convolutions with batch normalization layers. The regression head consists of two convolutions with ReLU layers. For each anchor on the regression head, we added an indirect supervision implicit objectness prediction for each anchor. With the corresponding implicit objectness, high response was suppressed in the background region. The final score was generated by multiplying the output of the classification head.

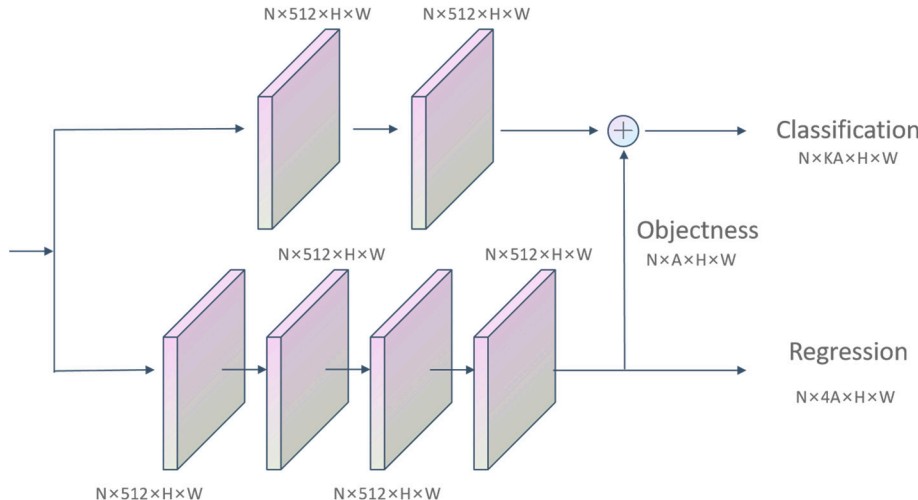

**Figure 8.** The classification and regression head with two and four convolution layers, respectively.

The method uses focal loss on the output of classification. The loss layer combines with the sigmoid operation for computing with the loss, which leads to greater stability. Simple samples and difficult samples can be clearly distinguished with focal loss. For simple samples, the corresponding focal loss value would be small. For difficult samples, the corresponding focal loss value would be large. The strengthen matching method compensates for the loss of difficult samples and makes it more equal for different samples. GIoU loss is used in regression instead of IoU loss. In addition to focusing on the overlapping area, GIoU also focuses on the non-overlapping area, which can better reflect the overlap degree.

For the decoder, we adopted two parallel heads, including classification head and regression head. We made the number of convolution layers in the two heads different. A strengthen matching method was used to make sure that all ground truth boxes can be matched and contribute equally to solve the imbalance problem with positive anchors.

## 4. Datasets

To validate our proposed method, we conducted an experiment on the DIOR [27] dataset and the NWPU-VHR-10 [28] dataset. This section describes the two remote sensing datasets in detail.

### 4.1. DIOR Dataset

DIOR [27] included 23,463 optimal remote sensing images with 20 common object categories in which 192,472 object instances were labeled. The spatial resolution of the images varied from 0.5 to 30 m. As with most of the existing datasets, DIOR was collected from Google Earth by Google company. The size of each image is 800 × 800 pixels. This dataset has the largest size and categories. Images in DIOR contain rich size variations for

the same category and the same category has different sizes due to spatial resolution. The characteristics of the dataset include larger image size, richer information, similarity within the class, and similarity outside the class. The train set and validation set of DIOR are set as 1:1. Examples of the dataset are shown below.

As shown in Figure 9, there are 20 object classes: airplane, airport, baseball field, basketball court, bridge, chimney, dam, expressway service area, expressway toll station, golf field, ground track field, harbor, overpass, ship, stadium, storage tank, tennis court, train station, vehicle, and windmill. Table 2 shows the detailed number of images per object class and per subset of the DIOR dataset.

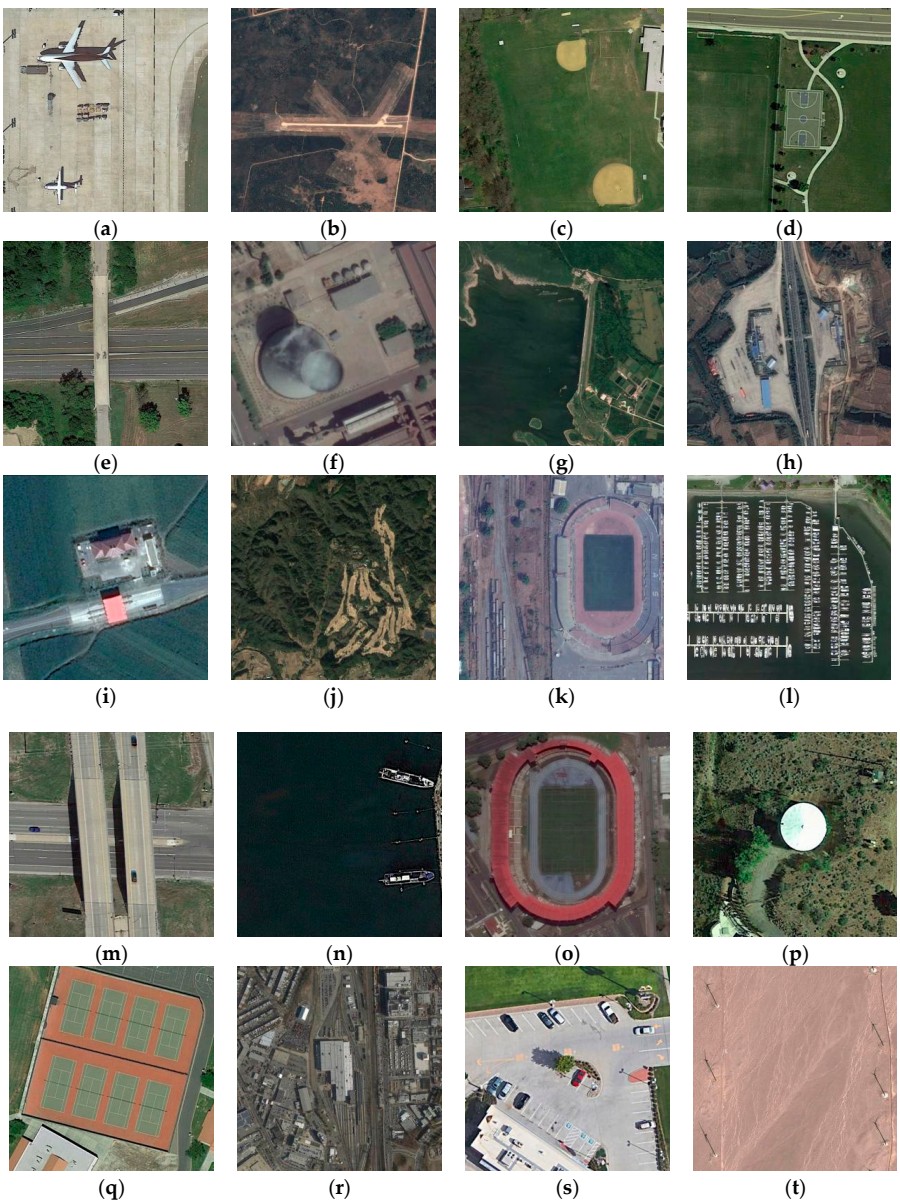

**Figure 9.** All 20 classes object (**a**–**t**) in DIOR dataset. (**a**) Airplane; (**b**) Airport; (**c**) Baseball field; (**d**) Basketball court; (**e**) Bridge; (**f**) Chimney; (**g**) Dam; (**h**) Expressway service area; (**i**) Expressway toll station; (**j**) Golf course; (**k**) Ground track field; (**l**) Harbor; (**m**) Overpass; (**n**) Ship; (**o**) Stadium; (**p**) Storage tank; (**q**) Tennis court; (**r**) Train station; (**s**) Vehicle; (**t**) Windmill.

**Table 2.** Details of instances number in DIOR dataset [27].

|  | Train | Val | Trainval | Test |
|---|---|---|---|---|
| Airplane | 344 | 338 | 682 | 705 |
| Airport | 326 | 327 | 653 | 657 |
| Baseball field | 551 | 577 | 1128 | 1312 |
| Basketball court | 336 | 329 | 665 | 704 |
| Bridge | 379 | 495 | 874 | 1302 |
| Chimney | 202 | 204 | 406 | 448 |
| Dam | 238 | 246 | 484 | 502 |
| Expressway service area | 279 | 281 | 560 | 565 |
| Expressway toll station | 285 | 299 | 584 | 634 |
| Golf course | 216 | 239 | 455 | 491 |
| Ground track field | 536 | 454 | 990 | 1322 |
| Harbor | 328 | 332 | 660 | 814 |
| Overpass | 410 | 510 | 920 | 1099 |
| Ship | 650 | 652 | 1302 | 1400 |
| Stadium | 286 | 292 | 851 | 619 |
| Storage tank | 391 | 384 | 775 | 839 |
| Tennis court | 605 | 630 | 1235 | 1347 |
| Train station | 244 | 249 | 493 | 501 |
| Vehicle | 1556 | 1558 | 3114 | 3306 |
| Wind mill | 404 | 403 | 807 | 809 |
| Total | 5862 | 5863 | 11,725 | 11,738 |

*4.2. NWPU-VHR-10 Dataset*

NWPU-VHR-10 [28] contains a total of 800 images, 650 images for train and 150 images for test, which were collected from Google Earth. It has 10 object classes and a total of 3775 instances. The spatial resolution of the images varied from 0.5 to 2 m. The dataset has been widely used for remote sensing object detection. The train set, validation set, and test set of NWPU-VHR-10 dataset are set as 7:2:1.

Object instances in NWPU-VHR-10 include 757 airplanes, 390 baseball diamonds, 159 basketball courts, 124 bridges, 224 harbors, 163 ground track fields, 302 ships, 655 storage tanks, 524 tennis courts, and 477 vehicles. Compared with DIOR, NWPU-VHR-10 is also a dataset collected by Google Earth, which has overlapping categories with DIOR. However, the number of categories is down by half, and the size and number of images are much smaller. We used it to verify the validity of the method that we proposed secondarily. Figure 10 shows images in the NWPU-VHR-10 dataset. In the figure, Figure 10a,b is a positive image including detection objects. Negative images Figure 10c,d are seen below without objects for detection.

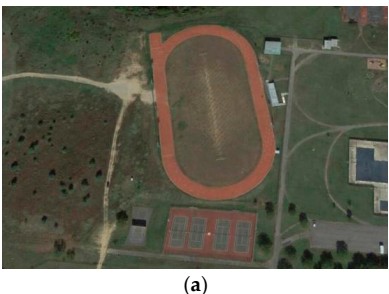
(**a**)

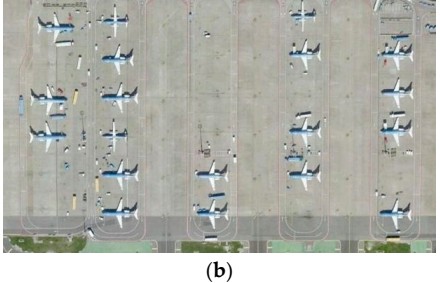
(**b**)

**Figure 10.** *Cont*.

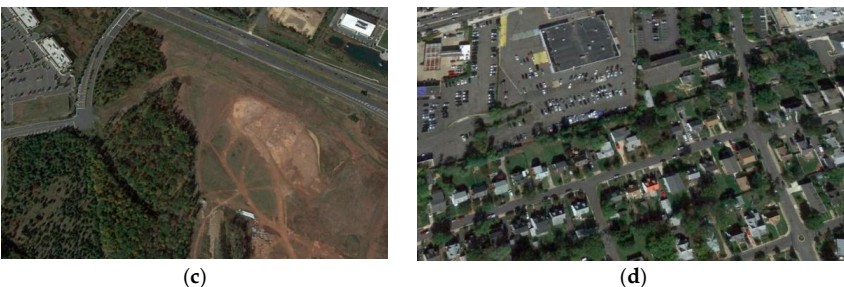

**Figure 10.** Remote sensing images in NWPU-VHR-10 dataset. (**a**,**b**) are positive pictures and negative (**c**,**d**) below.

## 5. Experiment

All the models were trained over NVIDIA 1080Ti GPU with 32 GB of RAM, based on Python 3.8. The experimental platform is Ubuntu 16.04 LTS. Other schedules follow the principles in Detectron2. Figure 11a–f show the visualization results for tests on the DIOR dataset and Figure 11g–l show results for tests on the NWPU-VHR-10 dataset with the StrMCsDet in this paper.

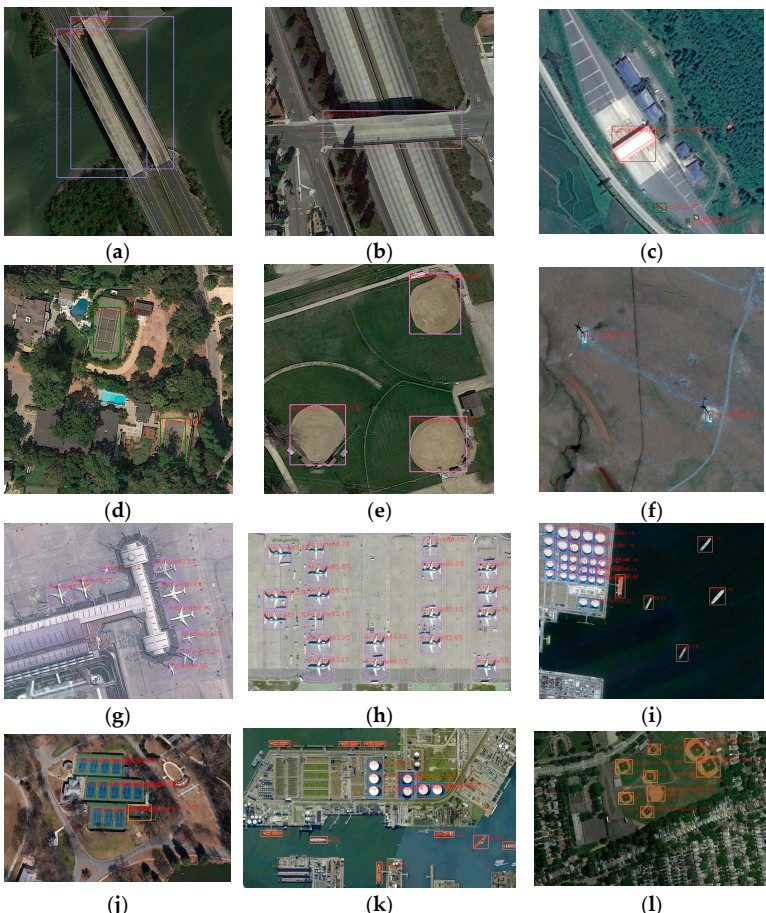

**Figure 11.** Visualization result of detection. (**a**–**f**) are test results on the DIOR dataset and (**g**–**l**) are test results on the NWPU-VHR-10 dataset.

### 5.1. Evaluation Metrics

In the object detection field, the performance of a detector often uses the average precision of each category and the mean average precision of all categories to evaluate. The precision indicates the model's ability to correctly detect the object while the recall shows the ability of the model to find the object. The calculation formulas for precision and recall

rate are as follows (1) and (2), where TP, FP, FN, TN means true positive, false positive, false negative, and true negative, respectively.

$$\text{Precision} = \frac{TP}{TP + FP} \tag{1}$$

$$\text{Recall} = \frac{TP}{TP + FN} \tag{2}$$

AP and mAP refer to average precision and mean average precision, which are also important in object detection. APS, APM, and APL refer to precision across scales including small, medium, and large, which means object areas less than 322, between 322 and 962, and larger than 962. AP50 and AP75 refer to average precision under different IoU values, AP at IoU = 50 and 75 which is the PASCAL VOC metric and strict metric, respectively. The higher the average accuracy, the better the detector performance.

$$AP = \int_{0}^{1} P(R)dR \tag{3}$$

$$mAP = \frac{1}{N_{cls}} \sum_{i=1}^{N_{cls}} AP_i \tag{4}$$

Frames Per Second (FPS) was used to evaluate the speed of object detection. The number of images that can be processed per second or the time needed to process an image to evaluate the detection speed, the shorter the time, the faster the speed of the detector. In this paper, AP, AP50, AP75 mAP, APS, APM, APL, model size, and FPS were used to evaluate the proposed method.

*5.2. Comparison Experiment*

There were 20 classes in the DIOR dataset. Each serial number class corresponds to Table 3. In addition, all the experimental results are without data augmentation or any tricks.

**Table 3.** The 20 object classes in DIOR dataset.

| 1 | 2 | 3 | 4 | 5 | 6 |
|---|---|---|---|---|---|
| Airplane | Airport | Bridge | Vehicle | Ship | Expressway Toll Station |
| 7 | 8 | 9 | 10 | 11 | 12 |
| Golf Field | Harbor | Chimney | Dam | Overpass | Stadium |
| 13 | 14 | 15 | 16 | 17 | 18 |
| Train Station | Storage Tank | Ground Track Field | Tennis Court | Expressway Service Area | Windmill |
| 19 | 20 | | | | |
| Basketball Court | Baseball Field | | | | |

In this article, we compared our method with six state-of-the-art methods and two remote sensing detection methods, including both two-stage and one-stage methods. Faster-RCNN [4], SSD [6], Faster-RCNN with FPN [16] with backbone of ResNet50 [14] and ResNet101, RetinaNet [15] with backbone of ResNet50 and ResNet101, Yolov3 [9], Yolov4 [10], M2Det [40], and CF2PN [41]. In addition, YoloV4 combines many tricks, however, our work is to propose a simple and efficient model, so to be fair, free bags of tricks are not considered in this paper.

To make a fair comparison in the DIOR dataset, we conducted experiments with backbone VGG-16 in Faster-RCNN, SSD, M2Det, and CF2PN, backbone resnet-50 and resnet-101 in FPN and RetinaNet, and backbone Darknet53 and Darknet53-tiny in Yolov3 and Yolov4, respectively. In StrMCsDet, Darknet53 with cross stage partial net was used as the backbone. All hyperparameters are consistent in Detectron2. We set the batch size as 6 during the training periods over single 1080Ti GPU. For the DIOR dataset, the input image size and patch size were set at $800 \times 800$ as the original image size. The initial learning rate of all models was 0.01. We extended the number of warm up iterations from 500 to 1500; the threshold of NMS was 0.75. The weights were pre-trained on ImageNet which were used to initialize the model parameters.

Additionally, in our method, P5 layers, which have sufficient context for detection, were frozen by default to allow for fair comparison with other approaches. Outputs of the backbone are in the C5 feature map which has 2048 channels and has set a downsample rate of 16. For fair comparison with other approaches, the default setting for batchnorm layers was frozen.

Figure 12 shows the prediction visualization results. Figure 12a–e is the prediction result on Faster-RCNN [4], Faster-RCNN with FPN [16] with backbone of ResNet101 [14], RetinaNet [15], Yolov4 [10], and proposed StrMCsDet in this paper, respectively. The proposed method can reduce wrong prediction numbers and show a better performance on small objects.

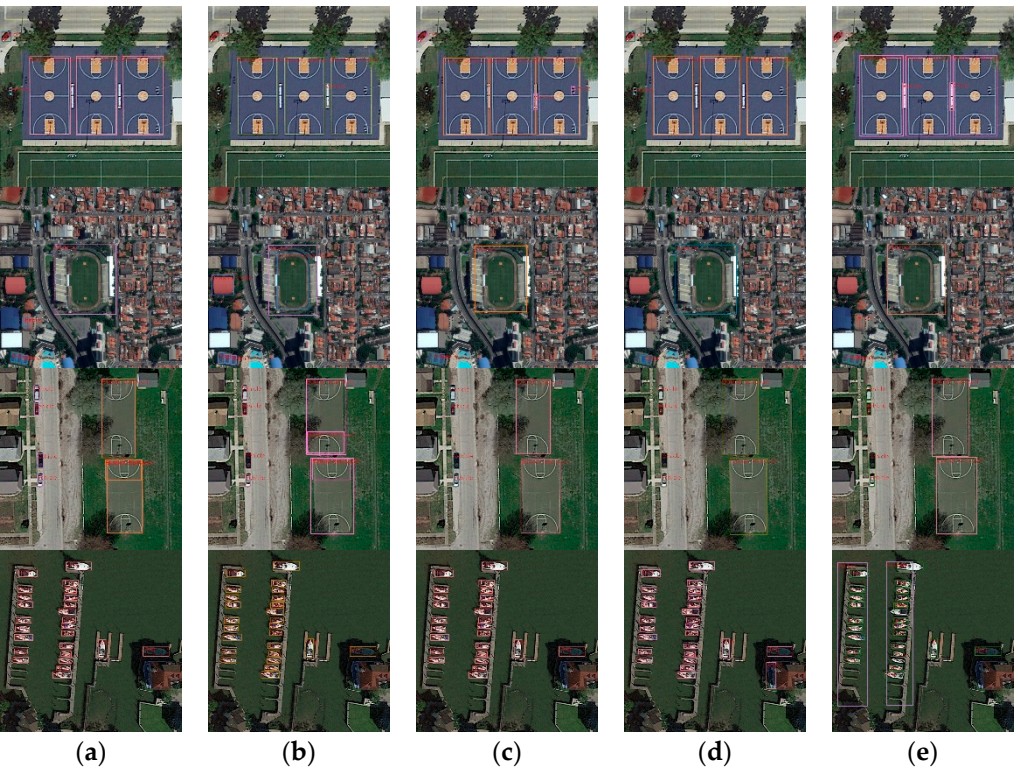

|  (a)  |  (b)  |  (c)  |  (d)  |  (e)  |

**Figure 12.** Prediction results comparison between Faster Rcnn, FPN, RetinaNet, YoloV4, and proposed method. (**a**) Faster Rcnn; (**b**) FPN; (**c**) RetinaNet; (**d**) YoloV4; (**e**) StrMCsDet.

Figure 13 shows the visualization comparison of detection results for small object tennis. The method proposed in this paper can effectively avoid missing small object detection. In comparison methods, tennis has been missed once or twice in prediction results. The proposed method detected all the tiny tennis courts in the image. In other images including tiny or small objects, the proposed method also shows a more accurate result than other methods.

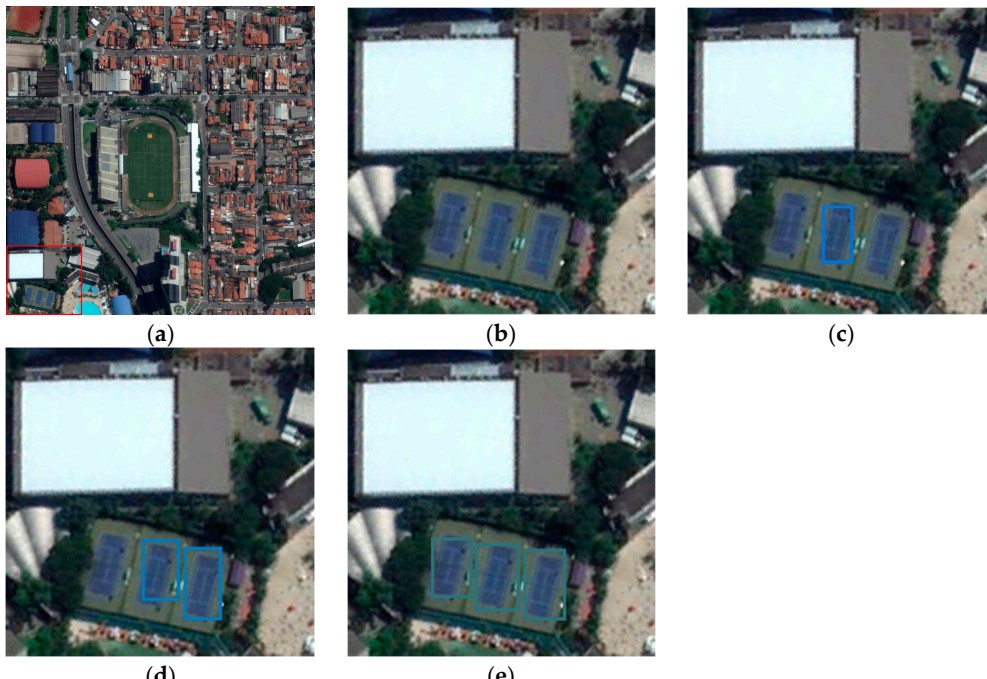

**Figure 13.** Details of detection results for small object of each methods. (**a**) is the original picture with the red box highlighted. Figure (**b**) shows local details of the original image in the red box. (**c**–**e**) are the predict results of YoloV4, RetinaNet, and proposed method, respectively. Tennis is one of the small object classes in the DIOR dataset. All boxes in (**c**–**e**) represent the visualization results of 'Tennis'.

The comparison of the proposed methods on the DIOR dataset is presented in Tables 4 and 5. The size means the amount of model parameters and the time means the average inference time of the model. All experiments models were trained in 56 k iterations.

**Table 4.** Comparison experiments of the AP values for different methods for the DIOR dataset, object class number correspond to Table 3.

| Methods | Backbone | C1 | C2 | C3 | C4 | C5 | C6 | C7 | C8 |
|---|---|---|---|---|---|---|---|---|---|
| Faster RCNN | VGG-16 | 43.67 | 49.35 | 28.09 | 23.62 | 27.74 | 55.25 | 55.60 | 30.23 |
| SSD | VGG-16 | 72.73 | 54.80 | 28.93 | 38.91 | 33.89 | 44.34 | 52.03 | 30.11 |
| FPN | ResNet-50 | 54.13 | 53.42 | 42.61 | 43.12 | 51.83 | 52.18 | 53.18 | 35.03 |
| | ResNet-101 | 54.02 | 54.54 | 44.84 | 43.17 | 51.81 | 52.39 | 56.02 | 38.49 |
| RetinaNet | ResNet-50 | 53.72 | 53.35 | 31.43 | 29.15 | 31.07 | 46.97 | 53.20 | 35.37 |
| | ResNet-101 | 53.45 | 54.37 | 30.22 | 28.73 | 31.33 | 47.83 | 52.82 | 35.46 |
| YoloV3 | Darknet53 | 67.09 | 59.71 | 38.35 | 31.82 | 59.67 | 52.40 | 50.93 | 30.65 |
| YoloV4-tiny | Darknet53-tiny | 59.22 | 65.02 | 41.58 | 32.99 | 47.12 | 46.41 | 56.26 | 30.32 |
| M2Det | VGG-16 | 68.03 | 58.34 | 33.54 | 34.17 | 39.87 | 43.55 | 52.49 | 31.50 |
| CF2PN | VGG-16 | 69.95 | 57.41 | 36.87 | 36.33 | 43.38 | 45.08 | 51.23 | 34.84 |
| Ours | CSPdarknet-C45 | 78.62 | 58.43 | 38.11 | 38.33 | 54.90 | 49.51 | 56.80 | 35.48 |
| | | **C9** | **C10** | **C11** | **C12** | **C13** | **C14** | **C15** | **C16** |
| Faster RCNN | VGG-16 | 50.97 | 62.34 | 50.15 | 43.07 | 38.66 | 39.81 | 56.92 | 35.23 |
| SSD | VGG-16 | 48.65 | 52.20 | 39.92 | 44.53 | 39.08 | 48.42 | 59.33 | 62.73 |
| FPN | ResNet-50 | 73.03 | 57.51 | 40.02 | 57.01 | 36.48 | 53.54 | 55.58 | 80.22 |
| | ResNet-101 | 72.55 | 60.08 | 42.23 | 68.31 | 39.51 | 53.58 | 56.86 | 79.87 |
| RetinaNet | ResNet-50 | 51.14 | 44.56 | 41.84 | 56.67 | 33.75 | 52.09 | 55.37 | 48.59 |
| | ResNet-101 | 50.24 | 44.72 | 42.35 | 56.31 | 33.32 | 52.33 | 57.84 | 48.31 |
| YoloV3 | Darknet53 | 65.17 | 54.68 | 39.88 | 45.03 | 38.34 | 33.79 | 59.67 | 34.64 |
| YoloV4-tiny | Darknet53-tiny | 47.25 | 67.33 | 40.10 | 48.47 | 40.98 | 29.60 | 59.68 | 28.43 |
| M2Det | VGG-16 | 69.19 | 46.48 | 39.74 | 49.87 | 36.54 | 43.85 | 54.78 | 49.56 |
| CF2PN | VGG-16 | 73.78 | 45.88 | 38.73 | 58.97 | 35.54 | 46.54 | 55.23 | 50.15 |
| Ours | CSPdarknet-C45 | 79.18 | 37.12 | 42.54 | 66.03 | 38.33 | 66.56 | 62.86 | 80.82 |

**Table 4.** *Cont.*

|  |  | C17 | C18 | C19 | C20 | mAP |
|---|---|---|---|---|---|---|
| Faster RCNN | VGG-16 | 49.03 | 45.48 | 36.22 | 48.80 | 48.83 |
| SSD | VGG-16 | 32.82 | 34.52 | 39.83 | 32.91 | 58.49 |
| FPN | ResNet-50 | 47.72 | 40.87 | 70.09 | 63.30 | 54.19 |
|  | ResNet-101 | 45.61 | 41.26 | 70.75 | 63.32 | 54.17 |
| RetinaNet | ResNet-50 | 37.92 | 41.31 | 69.04 | 78.08 | 61.07 |
|  | ResNet-101 | 37.74 | 41.51 | 69.02 | 78.19 | 60.35 |
| YoloV3 | Darknet53 | 49.19 | 31.72 | 37.91 | 79.02 | 57.96 |
| YoloV4-tiny | Darknet53-tiny | 40.91 | 31.90 | 37.76 | 79.41 | 55.53 |
| M2Det | VGG-16 | 46.65 | 36.60 | 69.56 | 77.45 | 56.21 |
| CF2PN | VGG-16 | 47.54 | 33.54 | 63.45 | 77.21 | 57.85 |
| Ours | CSPdarknet-C45 | 49.30 | 34.92 | 72.09 | 81.26 | 65.62 |

**Table 5.** Model size and inference speed of different methods for DIOR dataset on 1080Ti.

| Methods | Backbone | Size (M) | Time (ms) | FPS |
|---|---|---|---|---|
| Faster RCNN | VGG-16 | / | / | / |
| SSD | VGG-16 | 44.8 | 52.0 | 19.2 |
| FPN | ResNet-101 | 60.4 | 112.5 | 8.9 |
| RetinaNet | ResNet-101 | 45.9 | 102.5 | 9.8 |
| YoloV3 | Darknet53 | 54.3 | 51.0 | 19.6 |
| YoloV4-tiny | Darknet53-tiny | 50.7 | 47.4 | 21.1 |
| M2Det | VGG-16 | 59.6 | 39.2 | 19.8 |
| CF2PN | VGG-16 | 58.4 | 43.7 | 17.6 |
| Ours | CSPDarknetC5 | 41.4 | 26.0 | 38.5 |

Experiment results show that our method achieves the best performance on the mAP of 65.62 (%). Half of the 20 classes achieved the best accuracy compared with the other six SOTA methods. For the model size, our model only uses the C5 feature layer, which is the lightest model with 41.4 Mib, 19 ↓ than the FPN method. The inference time of our method is the fastest one with 26.0 ms and 38.5 Fps. In this work, the method we proposed demonstrated that utilizing the feature pyramid enough can make detection faster while using fewer feature maps, and the method can reach comparable results.

Similar to the DIOR dataset, we conducted experiments on the NWPU-VHR-10 dataset. The batch size was 8 and the input size was set at $800 \times 800$. We conducted experiments with backbone VGG-16 in Faster-RCNN, M2Det, and CF2PN, and backbone Darknet53 and Darknet53-tiny in Yolov3 and Yolov4, respectively. In StrMCsDet, Darknet53 with cross stage partial was used. All hyperparameters were consistent in Detectron2. The number of warm up iterations ranged from 500 to 1500; the threshold of NMS was 0.75. All the models were trained over NVIDIA 1080Ti GPU with 32 GB of RAM and 50 epochs.

The experimental results are shown in Table 6 Our method achieves the best performance of 75.22 mAP compared with Faster-RCNN [4], YoloV3 [9], and YoloV4 [10] on mAP, AP for different scales and IoUs. The result proved that our method is still able to achieve comparable results on different remote sensing datasets; the model of this paper has a generalized ability and robustness.

**Table 6.** Comparison experiments of the AP values for different methods for the NWPU-VHR-10 dataset.

| Methods | Backbone | mAP | AP50 | AP75 | APS | APM | APL |
|---------|----------|-----|------|------|-----|-----|-----|
| Faster RCNN | ResNet-101 | 57.43 | 91.78 | 62.81 | 40.97 | 56.69 | 61.16 |
| YoloV3 | Darknet53 | 65.51 | 92.45 | 59.53 | 41.43 | 57.16 | 66.83 |
| YoloV4-tiny | Darknet53-tiny | 63.27 | 91.34 | 58.77 | 43.45 | 59.83 | 64.32 |
| M2Det | VGG-16 | 67.10 | 93.62 | 61.20 | 40.10 | 61.02 | 72.53 |
| CF2PN | VGG-16 | 67.31 | 93.74 | 63.15 | 39.68 | 61.68 | 71.59 |
| Ours | CSPdarknetC5 | 73.53 | 95.05 | 69.04 | 46.64 | 68.38 | 75.22 |

*5.3. Ablation Study*

For the encoder part, residual blocks with dilated modules can provide gains to large objects and mildly improve the medium and small ones. More residual blocks bring more improvement, as shown in Table 7. The blocks' numbers represent the dilation of residual blocks in the encoder part. The experiment was conducted with the DIOR dataset. The large object was lightly affected by different dilation, but the small and medium objects were greatly affected. To make the model simple and accurate, we used 8 residual blocks by default. According to the comparison of experimental results, it is not suitable for the expansion convolutional encoder to carry out target detection on the image, and only 28.87 mAP can be obtained, that is, inaccurate detection results. With the increase of the number of blocks added to the encoder, the detection rate of the target has gradually improved in the three categories. For the cross-stage backbone proposed in this paper, blocks with a number of 8 achieve a balance between accuracy and speed.

**Table 7.** Ablation study for the DIOR dataset with the different encoder blocks. Blocks with 8 show a better result for remote sensing object detection.

| Blocks | mAP | AP50 | AP75 | APS | APM | APL |
|--------|-----|------|------|-----|-----|-----|
| 0 | 25.87 | 52.15 | 28.54 | 2.17 | 13.65 | 33.22 |
| 1 | 36.17 | 54.22 | 36.55 | 2.85 | 20.18 | 42.38 |
| 2 | 47.17 | 68.30 | 43.29 | 3.39 | 25.97 | 49.03 |
| 4 | 53.19 | 75.31 | 57.25 | 4.80 | 31.53 | 59.74 |
| 8 | 53.78 | 76.43 | 58.17 | 6.33 | 33.42 | 62.21 |

As shown in Table 8, numbers in the array represent details dilation. Residual blocks with channel and spatial attention modules have 0.78 ↑ and 0.75 ↑ with 4 and 8 blocks, respectively. This indicates that the light attention module has a certain positive effect in enhancing the accuracy of detection.

**Table 8.** Ablation study for the DIOR dataset. Effect with the different dilated block parameters and attention module.

| Dilated | CBAM | mAP | AP50 | AP75 | APS | APM | APL |
|---------|------|-----|------|------|-----|-----|-----|
| [1] [1] | | 42.17 | 62.20 | 44.05 | 3.90 | 23.18 | 51.38 |
| [1, 2, 3, 4] | | 53.19 | 73.31 | 57.25 | 4.80 | 31.53 | 62.24 |
| [1, 2, 3, 4] | √ | 53.97 | 75.31 | 57.84 | 6.16 | 31.79 | 63.58 |
| [1, 2, 3, 4, 5, 6, 7, 8] | | 53.78 | 76.43 | 58.17 | 6.33 | 33.42 | 62.51 |
| [1, 2, 3, 4, 5, 6, 7, 8] | √ | 54.53 | 76.68 | 59.24 | 6.79 | 33.40 | 64.15 |

[1] The numbers in square brackets represent the parameters of the dilated convolution. [1] represents one dilated encoder with original convolution layer while [1, 2] represents two dilated convolutions with parameters of 1 and 2.

The effect of strengthen matching and the dilated encoder with ResNet-101 proves that the encoder is an important part of a single-layer feature output. The original one-stage detector improved by the two components is shown in Table 9. The dilated blocks number

was set to 8. It performs worse on the original detector than our proposed method due to the design of the classification decoder. The ablation experiment proved that strengthen matching and the dilated encoder work positively in detection tasks, and are of a certain universality to improve the detector.

**Table 9.** Effect of strengthen matching.

| Strengthen Matching | Dilated | mAP | APS | APM | APL | ΔAP |
|---|---|---|---|---|---|---|
| | | 29.12 | 3.93 | 23.68 | 41.98 | 20.20 ↓ |
| | √ | 43.77 | 5.71 | 34.75 | 51.21 | 5.55 ↓ |
| √ | | 39.40 | 5.30 | 26.49 | 49.14 | 9.92 ↓ |
| √ | √ | 49.32 | 6.34 | 31.89 | 58.15 | - |

√ in the table represent use strengthen matching or dilated component. Method with both strengthen matching and dilated achieved the best preference. The difference between the controlled experiment and both used is indicated by ↓.

## 6. Discussion

In this work, we propose an efficient model for remote sensing object detection without using a full feature map. Accuracy can be achieved by using less training time and fewer computation resources on remote sensing datasets. Experiments on different public datasets demonstrate the universality of the method. Large objects can be detected more accurately and efficiently while small and medium objects are affected by dilated residual blocks that compensate for their limited receptive fields. However, AP for small and medium objects still has room for improvement under the remote sensing scenario. Additionally, the dynamic matching anchors may cause redundancy in very few images with crowded instances.

## 7. Conclusions

Nowadays, remote sensing images object detection methods are mainly focused on two-stage detectors with a feature pyramid structure and one-stage methods are based on single-shot detectors. However, complex feature pyramids slow down the detector and create computational burden. In this paper, the Cross-Stage Partial Strengthen Matching Detector (StrMCsDet) attempts to utilize single-feature output to replace the whole feature pyramid. The StrMCsDet achieves high speed and comparable precision in practical task-based remote sensing images. The method can deal with small objects in the real world without the feature pyramid structure, which provides a new baseline for one-stage object detection.

In this paper, we propose an efficient one-stage detector for remote sensing image object detection. A single-level feature layer is used without the full pyramid structure to fully utilize the feature map with a cross stage partial network backbone. We propose a line encoder with dilated blocks to compensate for the receptive field with the limited feature map and lightweight attention module to enhance the detector's performance. The accuracy and speed are comparable to those of the remote sensing dataset with the SOTA model despite having less computation. Additionally, the dynamic anchor operation in the decoder called strengthen matching improves ground truth box matching to a certain extent in order to address the imbalance and multi-scale problems. Experiments were conducted on the DIOR and NWPU-VHR-10 remote sensing image datasets. The experimental results show that our method is comparable in accuracy and has an excellent performance when applied to real world tasks.

Although the StrMCsDet achieved the most advanced results, it's performance on small objects in real world scenarios still has room to improve. One-level feature detectors provide a new baseline for remote sensing object detection to relieve computation burden. In future, lightweight, weather, light, and other influencing factors are still challenging for the algorithm. Meeting people's needs efficiently and accurately in real tasks is the direction that still needs to be explored.

**Author Contributions:** S.R. and X.G. conceived the idea; X.G. verified the idea and designed the study; S.R. and Z.F. analyzed the experimental results; X.G. gave comments on and suggestions for the manuscript. All authors have read and agreed to the published version of the manuscript.

**Funding:** This work was supported in part by the National Natural Science Foundation of China (42001408, 61806097), National Key R&D Program of China (2022YFE0204600).

**Conflicts of Interest:** The authors declare no conflict of interest.

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
