# Peer review of "A Cross Stage Partial Network with Strengthen Matching Detector for Remote Sensing Object Detection"

_remotesensing, doi:10.3390/rs15061574_

Round 1
Reviewer 1 Report
This paper presents a highly efficient one-stage object detector for remote sensing images. The proposed approach employs a single-level feature layer, which is different from the conventional full pyramid structure, and utilizes a cross-stage partial network backbone to fully exploit the feature map. The approach achieves comparable accuracy and speed on remote sensing datasets while requiring less computation, thus making it an attractive solution for practical applications. However, there are still the following problems in the paper.
1. some pictures are not explained in the text.
2. the red letters marked on Figure 13 are difficult to identify.
3. the algorithms compared in this paper are some general detection algorithms and not compared with the special detection algorithm Ref [23-26] in remote sensing.
4. it is suggested to explain the characteristics of this algorithm in remote sensing.
Author Response
Thank you for your comments. We agree with the comment and update the text and picture in the revised manuscript as the following:
- We have made more detailed explanations for the pictures, and some pictures have been changed, as shown in Fig.3,5,6.
- In Fig.13, the red letters have been removed, which represent object ‘Tennis’. The overall picture and the local picture (a) and (b) have been added. Since there is only one object category in the detail display of the figure, the red letters are removed and explained in the caption.
- We have added some methods combined with FPN (Feature Pyramid Network) structure designed for remote sensing in the experimental part for comparison, including ref [40]-[41]. Some recent related work added in part 3.
- We have revised the abstract, related work and conclusion, and further explained the characteristics of this algorithm in remote sensing..
Reviewer 2 Report
The author has proposed Cross Stage Partial Strengthen Matching Detector (StrM- 11 CsDet). The StrMCsDet generates a single-level feature map architecture in the backbone with cross stage partial network. The sound of this paper is quite interesting. Experiments on several datasets verify the effectiveness of the proposed method. I have the following concerns:
1. The abstract is not very clear; therefore, the abstract should be revised as per the theme of this paper.
2. What is the contribution of the manuscript? The proposed is the combination of several existing techniques, CNN based have proposed, and these methods are well exploited in the remote sensing domain. What difference is the proposed method from other existing ones?
3. The related work has some old ref, the author updated the ref with recent year publication and suggested to look the following ref to cite in his paper
· https://doi.org/10.1155/2022/9851533
· doi: 10.1109/LGRS.2022.3229556
· doi: 10.1109/TNNLS.2021.3105484
· doi: 10.1007/s10291-021-01181-4
· https://doi.org/10.21278/brod73102
· doi: 10.1016/j.compeleceng.2022.107685
· doi: 10.1007/s10489-021-03121-8
· doi: 10.1016/j.apt.2021.08.038
· doi: 10.1111/mice.12674
· doi: 10.3837/tiis.2022.01.001
· DOI: 10.3389/fmars.2022.1086140
4. So many abbreviations are used in the entire paper that, again, it is very difficult to understand. It should be double-checked.
5. Hyperparameters have a significant impact on machine learning methods. The authors should add a new part in the paper to discuss how to choose the hyperparameters for their method or other deep learning models, such as R-CNN.
6. Clearly explain the research problem or the challenges for object detection the paper is addressing.
7. It is suggested to rewrite the conclusion as per the main theme of the paper. it will be worthy if add future work.
8. The figures (Fig.1,2,3,5,6,7,8, 9, 10, 11, 12, 13) used in the manuscript, have poor resolution, and highly suffer to provide understanding about the concept and results.
9. The format of the literature citation is not uniform and needs to be adjusted.
10. the plagiarism should be reduce.
11. The manuscript requires English polishing. You need to conduct a thorough edit, addressing technical errors of grammar and wording as well as general readability and style.
Author Response
Thank you for your comments. We agree with the comment and update the text and picture in the revised manuscript as the following:
- We revised the abstract, and further explained the problem of remote sensing image detection.
- We revised the related work and conclusion. Some related works in remote sensing target detection are added, and comparison is added in the experiment, such as ref [40]-[41].
- We cited most of the recommended suitable articles and checked the article citations.
- We checked the abbreviations of the full text and added the full name of the noun when it first appeared.
- In 5.2 experiment, detailed instructions are added to supplement the experimental methods.
- In the abstract, related work section, the challenges encountered in the field of remote sensing target detection are explained, and new papers are cited to illustrate the difference between the use of single-layer features and the use of complete pyramids.
- In the conclusion, we added explanation of the research problem and the challenges for remote sensing object detection. Future work in conclusion and directions for further research added.
- The figure used in the paper has been modified, the image of remote sensing image has been replaced with the original high-resolution image. Some method pictures have been modified, and explanations have been added to the subtitles and the text of the article.
- We apologize for the language problems in the original manuscript. We fixed problems with citations, formatting, checking, and polishing the language to make it easier to read.
Reviewer 3 Report
This Paper has proposed a cross stage partial network for detection, it has merits but still need to improve:
1、Figure 2 should improve its quality.
2、More relative references should be cited
3、There are many researches on these field, what problems are the paper proposed to resolve
4、Please state the effects of attention and encoder module
5、please add the discussion, comparisons, experiments and samples on different weather conditions
Author Response
Thank you for your comments. We agree with the comment and update the text and picture in the revised manuscript as the following:
- The figure used in the paper has been modified, the image of remote sensing image has been replaced with the original high-resolution image. Some method pictures have been modified, and explanations have been added to the subtitles and the text of the article.
- We cited more recent articles related and checked the article citations.
- We have revised the abstract, conclusion, and explained the difference between the output feature with the single-layer pyramid and other methods. It is emphasized that this method can be used in remote sensing to make the detection light, efficient and accurate.
- In part 3.3, we have modified the body part to explain the effects of the encoder. Figure 6 has been updated to more clearly illustrate the structure of each block containing the dilation and attention modules.
- Some related works in remote sensing target detection are added, and comparison is added in the experiment, such as ref [40]-[41]. In the current datasets, different weather conditions are used for training and detection in the same dataset. In the conclusion part, we add future work and discuss the directions for further research.
Reviewer 4 Report
This paper proposes an object detector suitable for remote sensing applications. The main argument for the new architecture is that the current successful object detectors consider features at different levels something which involves more processing and hence leads to longer detection time (i.e. lower frames per second- FPS values). The proposed architecture, on the other hand, uses features only at a single level and hence achieves higher FPS values.
Strengths:
1. The paper proposes a novel architecture which is a based on features at a single level
2. Authors evaluate the system experimentally and the results indicate that the proposed system works better than some popular state-of-the-art systems such as Yolo and Faster-RCNN, with respect to both FPS and detection performance measured in mAP (mean average precision)
Weaknesses:
1. Some occasional English language and style errors can be found.
2. Presentation of the proposed method is generally poor, even though some parts of the presentation contain enough details and clear. Specific parts that need to be improved are
a) Figure 3(c): It is unclear how and where this fits into the other parts of the architecture (i.e.Figure 3(a) and 3(b).) It is not clear what the labels (eg: Transition, Part 1, Part 2 ) used in figure 3(c) refer to .
b) Figure 5: It is unclear how to interpret this figure. What is the meaning of the arrow? How a scale is represented by a circle (is it its size or position)? A clearer description is needed.
c) Figure 6: It is hard to comprehend the connection between Figure 6 and the text which describes it. Where are the dilated convolutions? Where is the attention operation?
d) Text between lines 305-312: This paragraph describes a key property of the method, it is recommended to have a figure to support this description.
Author Response
Thank you for your comments. We agree with the comment and update the text and picture in the revised manuscript as the following:
- We fixed problems with citations, formatting, checking, and polishing the language to make it easier to read.
- 3 has been updated to more clearly illustrate. Fig.3(a) is the backbone of the network. (c) is one of the cross stage partial stage in the backbone. We explain transition more detail in article below.
- We add explanation for Fig.5. Circle number represents the scale of object across the scale, the more the circle, the larger the scale.
- 6 has been updated to more clearly illustrate the structure of each block containing the dilation and attention modules.
- We added table.1 to describe the anchor size and aspect ratio of traditional one-stage detector with multi-layer output RetinaNet and the proposed method.
Round 2
Reviewer 2 Report
the author has answered all of my questions, this paper can be published in RS journal